# Screening of Antifungal Activity of Essential Oils in Controlling Biocontamination of Historical Papers in Archives

**DOI:** 10.3390/antibiotics12010103

**Published:** 2023-01-06

**Authors:** Ana Tomić, Olja Šovljanski, Višnja Nikolić, Lato Pezo, Milica Aćimović, Mirjana Cvetković, Jovana Stanojev, Nebojša Kuzmanović, Siniša Markov

**Affiliations:** 1Faculty of Technology Novi Sad, University of Novi Sad, Bulevar Cara Lazara 1, 21000 Novi Sad, Serbia; 2The Archives of Vojvodina, Žarka Vasiljevića 2A, 21000 Novi Sad, Serbia; 3Instutute of General and Physical Chemistry, Studenski trg 10-12, 11000 Belgrade, Serbia; 4Institute of Field and Vegetable Crops Novi Sad, University of Novi Sad, Maksima Gorkog 30, 21000 Novi Sad, Serbia; 5Institute of Chemistry, Technology and Metallurgy, University of Belgrade, Njegoševa 12, 11000 Belgrade, Serbia; 6Biosense Institute, University of Novi Sad, Dr Zorana Ðindića 1, 21000 Novi Sad, Serbia

**Keywords:** paper conservation, paper biodegradation, fungal contamination, essential oil application, fungicide effect

## Abstract

The main challenge in controlling the microbiological contamination of historical paper is finding an adequate method that includes the use of cost-effective, harmless, and non-toxic biocides whose effectiveness is maintained over time and without adverse effects on cultural heritage and human health. Therefore, this study demonstrated the possibility of using a non-invasive method of historical paper conservation based on plant essential oils (EOs) application. Evaluation of antimicrobial effects of different EOs (lemongrass, oregano, rosemary, peppermint, and eucalyptus) was conducted against *Cladosporium cladosporoides*, *Aspergillus fumigatus*, and *Penicillium chrysogenum*, which are commonly found on archive papers. Using a mixture of oregano, lemongrass and peppermint in ratio 1:1:1, the lower minimal inhibition concentration (0.78%) and better efficiency during a vapour test at the highest tested distance (5.5 cm) compared with individual EOs was proven. At the final step, this EOs mixture was used in the in situ conservation of historical paper samples obtained from the Archives of Vojvodina. According to the SEM imaging, the applied EOs mixture demonstrates complete efficiency in the inhibition of fungi colonization of archive papers, since fungal growth was not observed on samples, unlike the control samples.

## 1. Introduction

Nowadays, a great interest in the investigation of using essential oils (EOs) in diverse applications has emerged. This came about as a consequence of the high antimicrobial and antioxidant potential of EOs, so they could be used as a possible replacement for antibiotics and fungicides. EOs are commonly used in cosmetics and perfumes, and the food industry, but also have medicinal applications due to their therapeutic properties [1]. Taking into account the power of EOs, the newest research works are based on their alternative applications, such as part of smart packaging systems or as agents for controlling biocontamination in archives and libraries [2,3].

The prevention and treatment of biological degradation of paper is one of the biggest concerns in the field of preservation of paper materials of archival, library and museum collections. The combination of a high degree of biodegradation of paper with inappropriate microclimate conditions of storage and preservation make this material very sensitive to mould development [4]. In recent decades, much attention in the scientific research community has been focused on the evaluation of antimicrobial substances and testing methods of different materials depending on the characteristics of the material, the type and method of application, and the method of disposal/use [5,6,7,8]. The main challenges in the control of microbiological contamination of paper are reflected in finding methods and procedures that include the application of economically profitable biocides that contain harmless and non-toxic active compounds, and whose effectiveness is maintained over time and without harmful effects on cultural heritage and human health [7,8]. Due to their antimicrobial properties, well known since ancient times, natural molecules in plant essential oils are applied to suppress biological colonization. Essential oils contain a spectrum of plant secondary metabolites that can inhibit mould growth [9,10]. They act directly on the microbial cell by inhibiting its growth, causing ruptures of the cytoplasmic membrane, regulating intermediate metabolism, inhibiting enzymatic reactions, etc. [9]. On the other hand, their relationship to paper material is characterized by a high degree of inertness. Due to their highly effective antimicrobial activity against moulds, essential oils represent a potential ecological and economically viable solution for the biodegradation of archival paper materials [4,6,7].

Throughout history, several methods have been developed to prevent and stop the deterioration of paper-based archival materials that represent a significant cultural heritage of a country (e.g., birth records, death records, original historical records, unique and church records, clerical records, etc.) [11]. Most chemical and chemical–physical conservation procedures for the primary prevention of microbiological colonization are based on highly toxic compounds such as benzalkonium chloride, ethylene oxide, permethrin, sodium fluoride, etc. [5,11,12]. These compounds are also characterized by potential carcinogenicity, low degree of degradability, the possibility of uncontrolled pollution in the application environment, but also a minimal conservation effect in cases of already microbiologically contaminated samples of paper materials, which represents a big problem both for operators/conservators and for the treated material [12]. Additionally, most of these compounds are associated with shortcomings in application resulting from insufficient testing of their antimicrobial effect [5,13]. Moreover, there is increasing concern about the environmental and health issues of the use of toxic chemical agents in current conservation treatments. All the above-mentioned facts have led to the necessity of antimicrobial alternatives with lower toxicity as a long-term solution to improve the conservation and protection of historically important papers. Several scientific papers deal with using essential oils as alternative substances for antimicrobial treatment of historical papers [3,4,14,15,16], indicating the potential of these natural oils in the prevention and inhibition of fungal colonization of paper-based cultural heritage samples.

Based on the above facts, the aim of this paper is to investigate the possibility of using different EOs to control fungal contamination of historical paper. In order to improve further conservation and protection steps of archival materials, a comprehensive approach of selection and application of alternative biocides such as essential oils against archive-originated fungi was involved in this paper. To our knowledge, this is the first study that combines mathematical modelling with microbiological and chemical analysis to obtain the most efficient application method of EOs in historical paper treatment.

## 2. Results

### 2.1. Screening of the EOs’ Antifungal Activity

The antifungal activity of five EOs against three representative fungi commonly found on archive papers was evaluated by diameters of the inhibition zone by using disc diffusion method. As shown in Table 1, the highest antifungal activity against all tested strains was observed for oregano, lemongrass and peppermint EOs. The activity of these EOs was significantly higher than the activity of actidion, which was used as a positive control. On the other hand, rosemary and eucalyptus EOs exhibit moderate activity against *C. cladosporoides* (5/1), while no antifungal activity was observed against *A. fumigatus* (6/1) and *P. chrysogenum* (7/1). In this step, the inactivity of 5% DMSO against the tested fungi was confirmed, so it can be used for the preparation of lower concentrations of EOs.

### 2.2. Minimal Inhibitory Concentration Determination

Further testing of the antimicrobial effect against selected fungal representatives included a determination of minimal inhibitory concentrations (MICs) of EOs which have previously showed antifungal activity. According to the obtained results for individual essential oils (Table 2), MICs ranged between 0.78 and 6.75% of the initial concentration of EOs. Additionally, a mixture of efficient EOs (MIX) was tested and the obtained values of MIC of each EO was lower (0.78% for each EO) compared with individual MICs of tested EOs, which indicates a synergistic antifungal effect of tested EOs.

### 2.3. Time-Kill Kinetics Study

The time-kill kinetics profile of essential oils was determined against selected fungi using MICs of individual essential oil as well as the mixture of all three together in its MIC values (marked as MIX). Figure 1 shows kinetic models for control samples (non-treated fungal suspensions) and time-kill kinetics models for treated suspensions.

Additionally, Table 3 summarizes the regression coefficients of the observed control and time-kill kinetics models to clarify the speed and intensity of the used concentration of essential oils.

The goodness of fit between experimental quantities and model planned results are presented in Table 4. The gained kinetics models can be described as accurate since the coefficient of determination is high (*r*^2^ > 0.938). According to the time-kill kinetic models (Figure 1, Table 4), statistically calculated parameters fitted in well with obtained experimental data, with coefficients of determination between 0.938 and 0.998.

### 2.4. Chemical Composition of Selected EOs

The main compounds detected in three effective essential oils are presented in Table 5, while complete chemical compositions were shown in Appendix A. According to the gained results, the most abundant compounds in oregano essential oil (66 compounds, comprising 99.9%) were γ-terpinene (19.6%), carvacrol (15.6%), p-cymene (11.0%), and sabinene (8.8%). Menthol (30.3%), menthone (23.8%), menthouran (7.5%), and menthyl acetate (5.6%) were dominant in the chemical composition of peppermint essential oil (50 compounds, comprising 100%). In lemongrass essential oil (23 compounds, comprising 99.8%), geranial (51.5%) and neral (36.9%) were dominant followed by myrcene (4.8%) and geraniol (2.9%).

### 2.5. EOs Efficiency in Vapour Phase

After validation of high antifungal activity of the chosen three EOs in direct contact with fungal isolates, the efficiency of the EOs’ vapour phase on the fungal growth was evaluated using predictive mathematical analysis based on the surface area of fungal growth on Petri plates. Different distances between EO samples and inoculated Petri plates (1–5.5 cm) were analysed. The obtained images of Petri plates were presented in Appendix A. The obtained data were subjected to computer determination of the contaminated area, and the obtained surface area values were presented in Table 6. The complete absent of fungal growth was defined as 0% (negative result), while the fungal growth on the entire plate surface was determined as 100% (positive result). All presented results are quantified between these two values.

The obtained results of fungal growth on Petri plates were evaluated through regression coefficients, as presented in Table 7. The calculation was carried out using exponential model analysis based on Equation (3). Due to the absence of growth in the case of *C. cladosporoides* (5/1) for all essential oils and the mix, a regression coefficient cannot be determined (Table 8). The same situation was observed for *P. chrysogenum* (7/1) in the case of using a mixture of essential oils.

As the next step in the evaluation of EOs, efficiency in vapour phase, the predictive modelling was conducted. The goodness of fit between experimental quantities and model planned results are presented in Table 8. The gained kinetics models can be described as accurate since the coefficient of determination is high (*r*^2^ > 0.721). According to the gained models, statistically calculated parameters fitted in well with obtained experimental data, with coefficients of determination between 0.721 and 0.985.

### 2.6. EOs In Situ Application onto Historical Paper Samples

In order to verify decent results provided in previous experiments, the efficacy of an EOs mixture in vapour phase was tested in situ on the historical paper samples. SEM analysis of the samples was carried out in order to detect the difference in microstructure of the historical paper samples artificially contaminated with fungi after the treatment with vapour of EOs mixture, and the same artificially contaminated samples that were not treated with vapour of EOs. In Figure 2a–f, the SEM micrographs for artificially contaminated historical paper substrates non-treated and treated with the vapour of EOs mixture are shown, while in Figure 2g the SEM micrograph of the paper which was used as a blank sample is presented. All of the micrographs were carried out at the magnification ×500.

## 3. Discussion

In recent decades, natural molecules in plant essential oils have been utilized for the suppression of biological colonization in/on different materials. The antimicrobial potential of essential oils is based on their wide range of secondary metabolites that can inhibit mould growth but also possess inert relationships to materials [9,10]. Therefore, the in vitro antimicrobial activity was verified through comprehensive testing, which included several steps: the screening of antifungal activity (Table 1) for five essential oils (lemongrass, oregano, rosemary, peppermint, and eucalyptus), the determination of MIC concentrations (Table 2) for EOs which have fungicide effect, as well as time-kill kinetics for gained MICs against all fungal representatives (Figure 1, Table 3 and Table 4).

According to the gained results of the disc-diffusion method, the EOs of oregano, lemongrass and peppermint showed significant antifungal activity against all three tested fungi representatives. The wide inhibition zones with values greater than 38 mm for all tested fungi imply the possibility of using these EOs as fungicide agents. These results are in correlation with the results obtained by other researchers for peppermint [17,18,19,20], lemongrass [21,22,23,24] and oregano EOs [25,26,27]. The EOs of the mentioned plants had relatively high antimicrobial activities against many microorganisms, but especially against eukaryotic microorganisms such as the test organisms in this study. The average antimicrobial effect was detected in the case of rosemary and eucalyptus EOs against *C. cladosporoides* (inhibition zones were 21.33 and 14.67 mm, respectively), while the antimicrobial activity against *A. fumigatus* and *P. chrysogenum* were not detected. Based on these results, in can be concluded that *Cladosporium* representative is the most sensitive tested microorganism, while *Aspergillus* and *Penicillium* have similar resistant potential. Considering the gained results, oregano, lemongrass and peppermint EOs were chosen as the most potent EOs for further experiments. In the following test, the MIC values for selected EOs were determined. The obtained results have confirmed that the most sensitive fungus was *C. cladosporoides*, since the concentration of 1.56% was sufficient for the fungicide effect in the case of lemongrass and peppermint EOs, and 0.78% when Oregano EO was applied. For a fungicide effect against *A. fumigatus* and *P. chrysogenum*, the same MICs of 6.75% and 1.56% were determined for lemongrass and oregano, respectively. Only in the case of peppermint EO were the MIC values different, implying a four times higher concentration against *A. fumigatus* (Table 2). Additionally, a fungicide effect against all three fungal representatives can be achieved with lower values of the mentioned EOs (0.78%), using the MIX where the synergistic effect of EOs was accomplished.

In addition to antimicrobial profiling of the selected essential oil, the time-kill kinetics study was conducted. Sim et al. [28] emphasized that monitoring the in vitro pharmacodynamics potential of plant based-substances through a time-kill kinetics study can be a basis for understanding the biocide effect of MIC concentration during contact time. Therefore, the gained results are presented in Figure 1 and Table 3 and Table 4. Briefly, the time-kill kinetics profiles for *C. cladosporoides* indicated the relatively rapid fungicide effect of the tested oils, showing the greatest results in the case of using a mixture of essential oils (MIX). A total reduction in fungal viability was observed after the first 36 h in the case of the MIX, while the same effect for the individual effect of essential oil was determined after 60 h. Time-kill kinetics profiles for A. *fumigatus* suggest a slight prolongation of time required for the fungicide effect of individual oils, as well as MIX. Briefly, the mentioned effect was achieved for 72 h in the case of individual oils, videlicet 48 h for the MIX. Comparing the obtained results for *C. cladosporoides* and *A. fumigatus*, it can be observed for *P. chrysogenum* during contact time with different oils and their MIX that the killing rates were higher and inactivation profile was more prompt in all tested cases. The fungicide effect was observed 12 h and 24 h earlier compared with the same effect in the case of contact of MIX and *C. cladosporoides* or *A. fumigatus,* respectively. The individual effect of essential oil on the total reduction of viability of *P. chrysogenum* can be observed in the first 48 h. It can be concluded that the MIX of essential oils accelerates the fungicide effect for all tested fungal isolates, so its use can be economically and time acceptable.

Based on a review of the scientifically relevant literature, different spectra of plant metabolites can be involved in a high antimicrobial response during contact with EOs. In this study, selected EOs were conducted to GC-MS analysis in order to determine chemical profiles and compositions. Comparing the referenced data from the literature about chemical composition with the obtained chemical compositions of EOs in this study (Table 5, Appendix A), it could be assumed that the largest number of *O. vulgare* essential oil do not have the same chemical composition compared with the presented oil in this study [29,30,31,32,33,34,35,36,37]. Briefly, the most abundant compound in this oil, γ-terpinene, was not defined in other studies as the most dominant compound. The most usual chemotypes are carvacrol, germacrene D, sabinene, β-ocimene, and β-caryophyllene chemotype. The reason for this variety in chemical composition can be based on different geographical origins, since the mentioned research was conducted in more than 20 different countries across the world. On the other hand, the essential oils *Mentha piperita* and *Cymbopogon citratus* have the usual chemical composition, less dependent on the geographical area of plant growth [20,21,22,23,24,38,39,40,41,42,43]. In summary, antifungal activity of the dominant compounds in all three tested essential oils was observed, especially γ-terpinene [25,26,27] and menthol [17,18,19]. As two dominant compounds which together represent citral, geranial (trans-citral) and neral (cis-citral) in *C. citratus* essential oil have also been studied as effective agents against different fungi in vitro [44]. It is known that many extrinsic and intrinsic factors greatly influence the composition of the EO and the concentration of major and minor components in its chemical profile. Additionally, there is limited information about the influence of individual compounds on antimicrobial properties. For example, it is proven that carvacrol damages the cell membrane of fungi by inhibiting the ergosterol synthesis, which constitutes the backbone of the fungal cell membrane [45]. On the other hand, in one experiment, the vapour of peppermint oil and two its main compounds (menthol and menthone) were tested toward antifungal efficiency against *Sclerotinia sclerotiorum* and *Mucor* spp. The result of this experiment indicates that menthol alone was found to be the compound that carries the antifungal properties of peppermint oil, whereas menthone alone did not show any action [46]. When it comes to geraniol and citral as main compounds in essential oils, it is found that they show signs of apoptosis mediated by DNA fragmentation in yeasts cells [47]

The efficiency of EOs’ vapour phase on the fungal growth was evaluated using EOs with the highest antifungal activity in preceding experimental steps. The natural vapouring capacity of EOs at ambient temperature was involved in this step, while distance between inoculated Petri plates and EOs source was varied between 1.0 and 5.5 cm. Monitoring of samples was carried out for 21 days and imaged using a digital camera (Appendix A). The fungal growth, i.e., contaminated areas at Petri plates, served for the determination of surface area using a computer quantification method. According to Table 6, distances between 1 and 3.5 cm were designated efficiency for all tested essential oils and the mixture in the case of all three fungi representatives. The first differences at fungal strain as well as distance level were observed at a distance of 4.5 cm between inoculated Petri plates and EO samples. Briefly, *C. cladosporoides* (5/1) did not grow regardless of applied essential oil sample. On the other hand, complete growth inhibition of *A. fumigatus* (6/1) was observed in the case of using oregano EO or EOs mixture (MIX), while contaminated areas of 32 and 43% were observed in the case of peppermint and lemongrass essential oils, respectively. A similar situation can be noticed for *P. chrysogenum* (7/1), since the contamination area of 15 and 54% was obtained for the same two mentioned essential oils as for *A. fumigatus* (6/1). A further increase of the distance to a value of 5.5 cm reduced the efficiency of oregano EO in the case of *A. fumigatus* (6/1) and *P. chrysogenum* (7/1), since contamination areas of 32 and 11% were observed, respectively. In summary, only the mixture of essential oils was completely efficient at all tested distances. These results were evaluated using mathematical modelling (Table 7 and Table 8), where the obtained predictive quality parameters point out that statistically calculated parameters fitted in well with obtained experimental data. The gained coefficients of determination are very high, and predictive capacity of the models can be used for following application of EOs mixture during in situ treatments.

The absent of fungal growth indicates the strong potential of a mixture of oregano, peppermint, and lemongrass EOs for the inhibition of fungal growth of *C. cladosporoides* (5/1), *A. fumigatus* (6/1) and *P. chrysogenum* (7/1) at a distance greater that 5 cm (Table 6, Table 7 and Table 8). In order to demonstrate the potential of the in situ application of the created EOs mixture, historical paper samples were treated with the EOs mixture and SEM imaging was conducted. In Figure 2a, the micrograph of *C. cladosporoides* (5/1) is presented. Analysing the microstructure, apart from the cellulose fibres, hyphae and spores originating from fungi are clearly distinguished. Spores are the roundish structures and hyphae are thinner threads, connected to the round spores. In Figure 2b, spores and hyphae of *A. fumigatus* (6/1) on the cellulose fibre matrix are clearly observed, as well as in Figure 2c for *P. chrysogenum* (7/1). In comparison to Figure 2a–c, in Figure 2d–f there are no observed spores nor hyphae structures. In addition, Figure 2d–f is very similar to Figure 2g, which represents the blank historical paper sample. In these figures, only the cellulose fibres and irregular residues left after the manual paper production can be observed. This confirms that the treatment with EOs mix in vapour phase was lethal for the *C. cladosporoides* (5/1), *A. fumigatus* (6/1) and *P. chrysogenum* inoculated on archive paper substrates.

## 4. Materials and Methods

### 4.1. Tested Fungi

In this study, the following three fungi are used: *Cladosporium cladosporoides*, *Aspergillus fumigatus* and *Penicillium chrysogenum*. Throughout the manuscript they are labelled as 5/1, 6/1 and 7/1, respectively (Figure 3). These strains belong to the private collection of the Laboratory of Microbiology, Faculty of Technology Novi Sad, University of Novi Sad. All three strains have been earlier reported as paper biodeteriogens [48]. The chosen fungi were maintained and cultured onto Potato Dextrose Agar (HiMedia, Mumbai, India). During culturing the inoculated medium was incubated at 25 °C for 7 days. Sterile Petri dishes, without venting, were used in all further experiments.

### 4.2. EOs

Five different commercial essential oils (lemongrass, oregano, rosemary, peppermint and eucalyptus) purchased from AlekPharm (Belgrade, Serbia), were utilized in the preliminary screening of antifungal activity. Information about product INCI and CAS numbers are given in Table 9, while detail chemical composition (GC-MS) is given in Appendix A.

### 4.3. Screening of the EOs Antifungal Activity

Five selected essential oils were utilized in the preliminary screening of antifungal activity by disc-diffusion method, which was completely performed according to Aćimović et al. [49]. Some important steps of this method imply the use of freshly prepared fungal culture (obtained as described in Section 4.1) for the inoculation of Sabouraud Dextrose Agar (HiMedia, Mumbai, India) and application of the EOs (15 μL) onto sterile discs in triplicate for each culture onto solidified media. As a positive control actidione (30 μg/mL) was used, while as a negative control sterile distilled water and 5% DMSO were chosen. After 7 days at 25 ± 2 °C, the halo zones around discs were measured using HiAntibiotic Zone Scale^TM^ (HiMedia, Mumbai, India). The obtained results of the antifungal activity of EOs were interpreted as follows: resistant—halo zone lower than 22 mm, intermediary effect—range of halo zone is 22–26 mm, and sensitive—halo zone above 26 mm.

### 4.4. MIC Determination

Minimal inhibitory concentration (*v/v*) was assessed for EOs that achieved the biggest halo zone in screening of their antifungal activity evaluated by disc-diffusion method. The decreasing concentrations of EOs (100−0.78%) were tested while determining the *MIC* value, following the procedure explained in detail by Micić et al. [50]. The dissolution of essential oils was performed by using 5% DMSO solution. The final *MIC* value was determined using Equation (1), in which the number of microorganisms in the positive control is labelled as *N_c_*, while *N_t_* stands for the number of microorganisms in the treated sample.
(1)$$MIC=\left(\frac{N_{C}-N_{t}}{N_{c}}\right)\cdot 100\left(\%\right)$$

### 4.5. Time-Kill Kinetics Study

For monitoring microbial behaviour during contact with MIC concentration of selected essential oils and fungi, the time-kinetics study was carried out using the procedure described by Aćimović et al. [51]. In brief, the contact time of the MIC concentration of EOs with fungal spore suspension (prepared in the nutrient medium using McFarland standard, spore suspension approx. 6 log CFU/mL) was conducted at 25 °C for 120 h, while the sampling time was in 0, 6, 12, 24, 36, 48, 60, 72, 84, 96, 108 and 120 h. The number of survival cells was determined after streaking of the sample aliquot onto SMA plates and incubation at 25 °C for 5 days. As well as tracking the effect of individual EOs, the mix of equal portion of three tested EOs was also monitored regarding the number of cells. The time-kill kinetics modelling was conducted to obtain the four-parameter sigmoidal numerical model which is suitable for biological systems [49], and which can be represented as an S-shaped model. The formula for modelling the mentioned system was presented as Equation (2), where (*y*(*t*)) is the number of viable/survival cells during the contact time between the MIC concentration of essential oil and cells, while *a*,*b*,*c* and d represent a minimum of the experimentally obtained values (*t = 0*), the Hill’s slope, the inflection point and the maximally obtained value (*t* = ∞), respectively.
(2)$$y\left(t\right)=d+\frac{a-d}{1+{\left(\frac{t}{c}\right)}^{b}}$$

### 4.6. Chemical Composition of Selected EOs

The selected essential oils used were exposed to GC-MS analysis with an Agilent 7890A apparatus equipped with a 5975 C MSD, FID, and an HP-5MS fused-silica capillary column (30 m × 0.25 mm, film thickness 0.25 μm). The carrier gas was helium, and its inlet pressure was 19.6 psi and linear velocity of 1 mL min^−1^ at 210 °C. The injector temperature was 250 °C, the injection volume was 1 μL, split ratio was 10:1. The components were identified based on their linear retention index relative to C8-C32 n-alkanes, in comparison with data reported in the literature (Adams4 and NIST17 databases). The relative percentage of the oil constituents were expressed as percentages by FID peak area normalization.

### 4.7. EOs Efficiency in Vapour Phase

For dominant fungal strains isolated from archive paper, the antifungal efficiency of the vapour phase of EOs which showed the highest antifungal activity in previous experimental steps was evaluated. For this purpose, standard disc-diffusion method as described in Section 4.2. was employed, with some modifications. Namely, sterile discs were not applied onto inoculated and solidified media, rather the disc was laid on the inside surface of the upper lid and 15 μL of individual 100% EO or their mix (5 μL of each EO) was placed onto the disc. The distance between the inoculated media and the upper lid was varied as follows: 1, 2.5, 3.5, 4.5 and 5.5 cm by setting the additional Petri dish circuit onto a previous one (Figure 4). As the negative control 15 μL of sterile distilled water instead of EOs was applied onto sterile disc, while actidione (30 μg/mL) was used as a positive control. The prepared Petri dishes were inverted on top of the lid and sealed with parafilm to prevent leakage of the vapour. The experiment was performed in triplicate.

The prepared groups of the samples were monitored at 25 °C for 21 days’ storage period. Each sample image was recorded with a common digital camera, which captured a region of roughly Ø90 mm (the macro function was used, for a more clear photograph). The obtained pictures were used to determine the contaminated area of the sample using ImageJ 1.53 r (used for masking the contaminated area) and Inkscape 1.0 (applied to measure the distances in pictures). The values of the recorded images’ surface area were assigned to the values between 0 and 1 (negative control images surface gained value 0, while positive control images surface was assumed to be 1).

The data are presented as an exponential model (Equation (3)).
(3)SE(t)=a·exp(b·t)

The surface area *SE*(*t*) at time *t* was the targeted output, while *a* and *b* were the regression coefficients. The suitability of the gained mathematical models was tested using the mathematical quality parameters through numerical verification of the obtained models according to Equations (4)–(7) [52,53].
(4)χ 2=∑i=1N(xexp,i−xpre,i)2N−n,
(5)$$RMSE={\left[\frac{1}{N}\cdot \sum_{i=1}^{N}{\left(x_{pre,i}-x_{exp,i}\right)}^{2}\right]}^{1/2},$$
(6)$$MBE=\frac{1}{N}\cdot \sum_{i=1}^{N}\left(x_{pre,i}-x_{exp,i}\right),$$
(7)$$MPE=\frac{100}{N}\cdot \overset{N}{\underset{i=1}{\sum }}\left(\frac{\left|x_{pre,i}-x_{exp,i}\right|}{x_{exp,i}}\right)$$

The *x_exp,i_* represents experimental values, while *x_pre,i_* are the predicted values calculated from the model. *N* and *n* are the numbers of observations and constants, respectively. In order to compare samples means, analysis of variance (ANOVA) was conducted.

### 4.8. EOs In Situ Application onto Historical Paper Samples

The samples for the in situ experiment were taken from a fragment of the handmade paper; it was impossible to determine to which document it belonged to while lacking text, seals and other insignia, and it is assumed to be a part of the fond F.2 Bačka-Bodrog County—Baja, Sombor, Serbia (1699–1849), 1688–1849. The date when this type of the paper was made is believed to be the end of the 18th and the beginning of the 19th century, at a time when the manufacturing of paper included the use of quality materials—fibre of more pure or less pure cellulose obtained from old cotton, linen and hemp rags, which had a certain amount of glue added. It is a strong, elastic and durable type of paper. Under optimal conditions for preservation, the properties of this type of paper do not experience significant changes over time, unlike the paper created by wooden materials. The paper is without visible damage, while a lesser discoloration (yellowing) is a consequence of the natural process of aging. The preparation of the samples included the cutting of the paper in equal parts with 1 × 1 cm dimensions.

Onto prepared historical paper samples 100 μL of prepared fungal suspensions (approx. spore concentration 6 log CFU/mL) was applied and placed in the bottom of the Petri dish, while the sterile disc with 15 μL of EOs mix (5 μL of each EO) or equal amount of distilled water for control samples was laid onto the inside of the upper lid. The distance between the artificially contaminated historical paper sample and soaked disc with mix of EOs was 3.5 cm. The experiment was carried out at 25 °C for 21 days. After three weeks, the state of the artificially contaminated and treated historical paper surface was evaluated by scanning electron microscopy (SEM Hitachi TM3030) under a high vacuum (acceleration voltage 15 kV, beam current 20 nA, spot size 1 mm).

## 5. Conclusions

The antifungal efficiency of three EOs (lemongrass, oregano and peppermint) against *Cladosporium cladosporoides*, *Aspergillus fumigatus* and *Penicillium chrysogenum* was demonstrated in liquid and vapour phase. Furthermore, through obtaining MIC values and time-kill kinetic study, the high antifungal potential of the tested EOs individually and in mixture have been shown. For the first time, mathematical modelling has been employed in order to track the efficacy of the EOs in vapour phase, where it was concluded that only a mixture of essential oils was completely efficient at all tested distances. Following the obtained results, the equal proportion of the tested oils was applied in situ onto small fragments of historical papers that had previously been artificially contaminated, and the same high efficiency of the EOs mixtures in vapour phase was confirmed by scanning electron microscopy.

After such promising results presented in this manuscript, the future plan of the authors is to accomplish a suitable protocol for the treatment of archival documents with EOs as well as to conducting experiments with the complete fonds of documents.

## Figures and Tables

**Figure 1 antibiotics-12-00103-f001:**
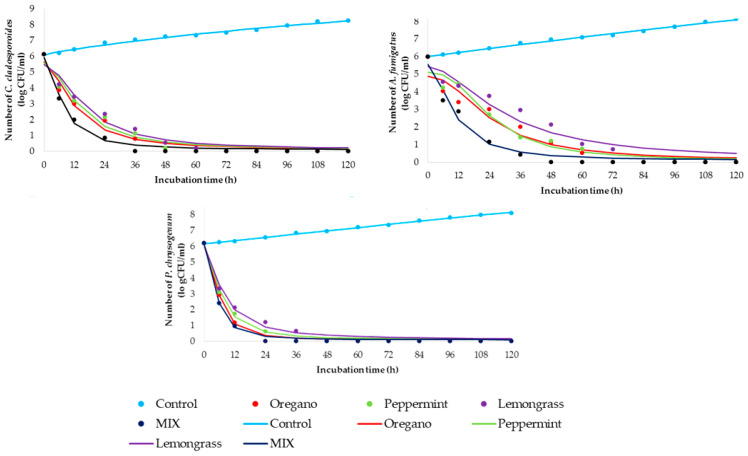
Time-kill kinetics study between MIC values of essential oils (markers signify the experimental data; lines indicate predictive results.

**Figure 2 antibiotics-12-00103-f002:**
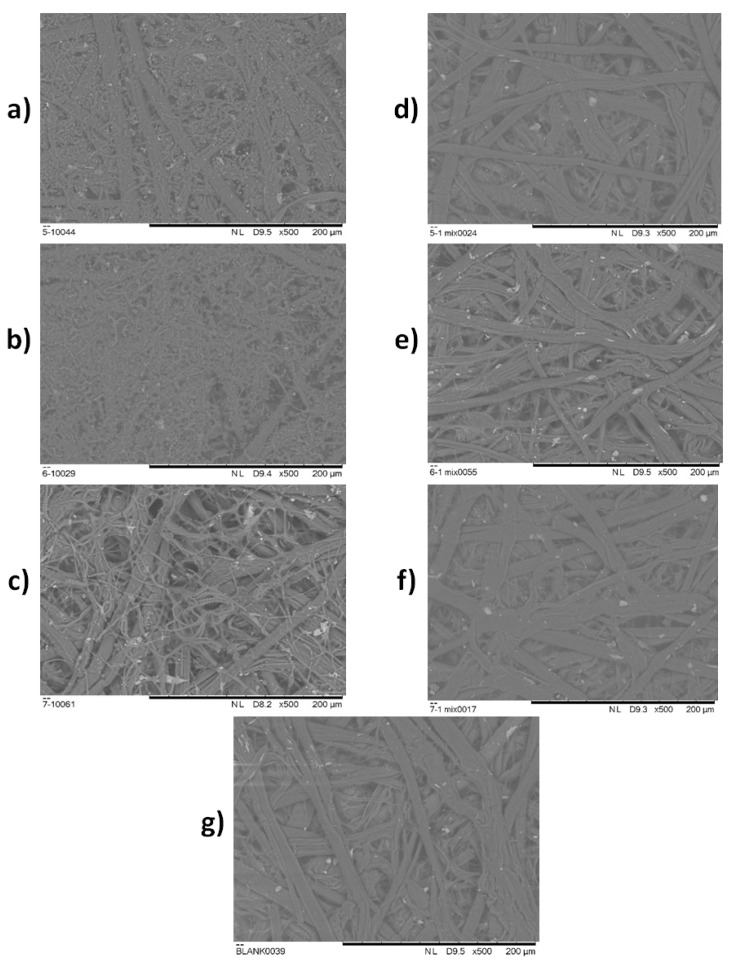
SEM micrographs for: (**a**–**c**) non treated samples *C. cladosporoides* (5/1), *A. fumigatus* (6/1), and *P. chrysogenum* (7/1), respectively; (**d**–**f**) treated samples *C. cladosporoides* (5/1) mix, *A. fumigatus* (6/1) mix and *P. chrysogenum* (7/1) mix, respectively; (**g**) blank historical paper sample.

**Figure 3 antibiotics-12-00103-f003:**
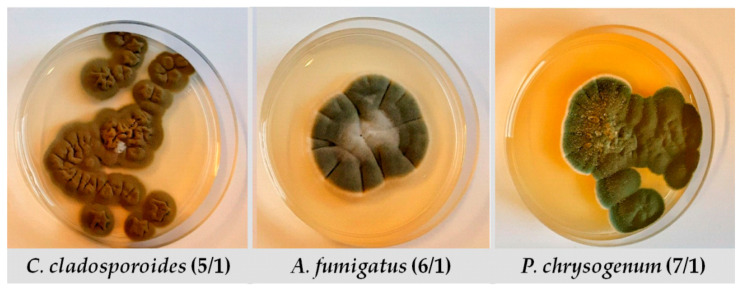
Tested fungal strains.

**Figure 4 antibiotics-12-00103-f004:**
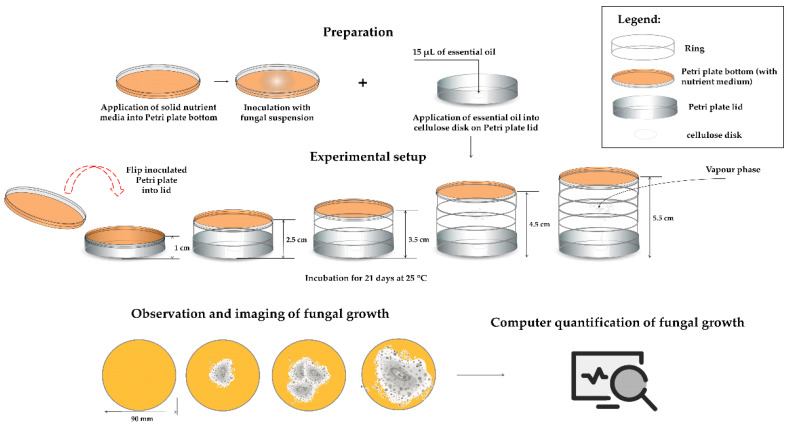
Schematic presentation of method for determination of EOs efficiency in vapour phase.

**Table 1 antibiotics-12-00103-t001:** Inhibition zones (mean value diameter (mm) ± standard deviation) observed on Sabouraud Dextrose Agar.

Essential Oil	Fungal Strain		
	*C. cladosporoides* (5/1)	*A. fumigatus* (6/1)	*P. chrysogenum* (7/1)
Lemongrass	39.33 ± 1.15	39.00 ± 1.00	38.33 ± 1.53
Oregano	40.00 ± 0.00	40.00 ± 0.00	40.00 ± 0.00
Rosemary	21.33 ± 3.46	nd *	nd
Peppermint	40.00 ± 0.00	39.33 ± 1.15	40.00 ± 0.00
Eucalyptus	14.67 ± 3.21	nd	nd
Actidion	26.67 ± 0.58	25.67 ± 0.58	22.20 ± 2.00
Sterile distilled water	nd	nd	nd
DMSO 5%	nd	nd	nd

* nd—not detected.

**Table 2 antibiotics-12-00103-t002:** The results of minimal inhibitory concentration (%) *** ± standard deviation.

Essential Oil	Fungal Strain		
	*C. cladosporoides* (5/1)	*A. fumigatus* (6/1)	*P. chrysogenum* (7/1)
Lemongrass	1.56 ± 0.00 ***	6.75 ± 0.00	6.75 ± 0.00
Oregano	0.78 ± 0.00	1.56 ± 0.00	1.56 ± 0.00
Peppermint	1.56 ± 0.00	6.75 ± 0.00	1.56 ± 0.00
MIX	0.78 ± 0.00	0.78 ± 0.00	0.78 ± 0.00

* the initial concentration of EO was determined as 100% and all lower concentration was obtained using the dilution method.

**Table 3 antibiotics-12-00103-t003:** Regression coefficients (a, b, c and d) for time-kill kinetics models (directed to Equation (2)).

Sample	Control	Oregano	Peppermint	Lemongrass	MIX
Fungal isolate	*C. cladosporoides* (5/1)				
d	16.261	0.100	0.100	0.100	0.100
a	6.081	5.642	5.592	5.493	5.925
c	549.710	12.062	13.871	16.368	7.337
b	0.862	1.811	1.852	1.890	1.838
	*A. fumigatus* (6/1)				
d	18.448	0.100	0.100	0.100	0.100
a	6.022	4.893	5.118	5.429	5.578
c	549.684	24.337	24.366	29.861	10.142
b	1.055	2.155	2.484	1.811	1.854
	*P. chrysogenum* (7/1)				
d	18.654	0.100	0.100	0.100	0.100
a	6.178	6.079	6.081	6.072	6.159
c	549.680	5.685	6.396	7.452	4.739
b	1.092	2.128	1.830	1.596	2.047

**Table 4 antibiotics-12-00103-t004:** The goodness of fit between experimental quantities and model planned results.

Fungi Isolate	Sample	Mathematical Model Quality Parameters									
		*χ* ^2^	RMSE	MBE	MPE	*r* ^2^	Skew	Kurt	Mean	StDev	Var
*C. cladosporoides* (5/1)	Control	0.006	0.076	0.002	0.927	0.988	−0.018	−1.33	0.0023	0.0795	0.006
	Oregano	0.148	0.369	−0.117	5.071	0.972	0.976	0.233	−0.117	0.365	0.133
	Peppermint	0.150	0.371	−0.113	20.223	0.972	1.074	0.008	−0.113	0.369	0.136
	Lemongrass	0.152	0.373	−0.103	20.307	0.971	1.06	−0.011	−0.103	0.374	0.140
	MIX	0.051	0.217	−0.109	3.268	0.992	0.754	−0.403	−0.11	0.195	0.038
*A. fumigatus* (6/1)	Control	0.003	0.057	0.001	0.642	0.993	−0.180	−0.566	0.001	0.059	0.003
	Oregano	0.287	0.513	−0.09	10.769	0.938	1.197	0.938	−0.09	0.528	0.279
	Peppermint	0.17	0.396	−0.061	7.400	0.965	0.86	1.821	−0.061	0.409	0.167
	Lemongrass	0.304	0.528	−0.146	11.658	0.949	0.459	−1.533	−0.1456	0.53	0.280
	MIX	0.109	0.317	−0.112	7.373	0.979	0.880	0.409	−0.112	0.309	0.096
*P. chrysogenum* (7/1)	Control	0.001	0.037	0.000	0.450	0.997	0.064	−1.202	−0.0003	0.038	0.001
	Oregano	0.029	0.163	−0.103	0.683	0.998	0.048	0.926	−0.103	0.132	0.017
	Peppermint	0.037	0.185	−0.108	2.001	0.995	0.881	0.407	−0.108	0.157	0.024
	Lemongrass	0.062	0.238	−0.109	4.752	0.989	0.741	−0.761	−0.109	0.221	0.049
	MIX	0.023	0.144	−0.103	0.754	0.998	0.0704	0.725	−0.103	0.105	0.011

Legend: *χ*^2^—reduced chi-square; RMSE—root mean square error; MBE—mean bias error; MPE—mean percentage error; *r*^2^—coefficient of determination; Skew—skewedness; Kurt—kurtosis; StDev—standard deviation; Var—variance.

**Table 5 antibiotics-12-00103-t005:** The dominant compounds in selected essential oil (complete chemical composition was given in Appendix A).

Essential Oil					
Oregano (*Origanum vulgare*)		Peppermint (*Mentha piperita*)		Lemongrass (*Cymbopogon citratus*)	
Compounds	%	Compounds	%	Compounds	%
γ-terpinene	19.6	Menthol	30.3	Geranial	51.5
Carvacrol	15.6	Menthone	23.8	Neral	36.9
p-cymene	11.0	Menthouran	7.5	Myrcene	4.8
Sabinene	8.8	Menthyl acetate	5.6	Geraniol	2.9

**Table 6 antibiotics-12-00103-t006:** Interpretation of data * for EOs efficiency in vapour phase based on contaminated surface area.

		Contaminated Area (%) *											
Microorganism		*C. cladosporoides* (5/1)				*A. fumigatus* (6/1)				*P. chrysogenum* (7/1)			
Essential Oil		Oregano	Peppermint	Lemongrass	MIX	Oregano	Peppermint	Lemongrass	MIX	Oregano	Peppermint	Lemongrass	MIX
Distance (cm)	1	0	0	0	0	0	0	0	0	0	0	0	0
	2.5	0	0	0	0	0	0	0	0	0	0	0	0
	3.5	0	0	0	0	0	0	0	0	0	0	0	0
	4.5	0	0	0	0	0	32	43	0	0	15	54	0
	5.5	0	0	0	0	32	53	61	0	11	91	92	0

* The complete absent of fungal growth was defined as zero (0)—negative result, while the fungal growth on the entire plate surface was determined as one (1)—positive result.

**Table 7 antibiotics-12-00103-t007:** Regression coefficients (a and b) for evaluation of EOs’ efficiency in vapour phase.

Regression Coefficients	Microorganisms											
	*C. cladosporoides* (5/1)				*A. fumigatus* (6/1)				*P. chrysogenum* (7/1)			
Essential Oil	Oregano	Peppermint	Lemongrass	MIX	Oregano	Peppermint	Lemongrass	MIX	Oregano	Peppermint	Lemongrass	MIX
a	-	-	-	-	0.676	0.897	0.821	0	0.496	1.233	0.912	-
b	-	-	-	-	0.003	0.004	0.007	0	0.004	0.001	0.006	-

**Table 8 antibiotics-12-00103-t008:** The “goodness of fit” between experimental quantities and model planned results.

Fungi Isolate	Sample	Mathematical Model Quality Parameters									
		*χ* ^2^	RMSE	MBE	MPE	*r* ^2^	Skew	Kurt	Mean	StDev	Var
*C. cladosporoides* (5/1)	Oregano	-	-	-	-	-	-	-	-	-	-
	Peppermint	-	-	-	-	-	-	-	-	-	-
	Lemongrass	-	-	-	-	-	-	-	-	-	-
	MIX	-	-	-	-	-	-	-	-	-	-
*A. fumigatus* (6/1)	Oregano	0.011	0.096	0.018	12.646	0.794	2.019	4.300	0.018	0.105	0.011
	Peppermint	0.005	0.062	−0.013	6.746	0.925	1.003	2.183	−0.013	0.068	0.005
	Lemongrass	0.010	0.091	−0.017	7.942	0.887	1.256	2.555	−0.017	0.100	0.010
	MIX	-	-	-	-	-	-	-	-	-	-
*P. chrysogenum* (7/1)	Oregano	0.001	0.030	−0.005	9.264	0.721	1.623	3.118	−0.005	0.033	0.001
	Peppermint	0.005	0.061	−0.035	14.865	0.985	−0.280	−1.523	−0.035	0.055	0.003
	Lemongrass	0.013	0.103	−0.022	6.539	0.931	0.952	2.119	−0.022	0.113	0.013
	MIX	-	-	-	-	-	-	-	-	-	-

Legend: *χ*^2^—reduced chi-square; RMSE—root mean square error; MBE—mean bias error; MPE—mean percentage error; *r*^2^—coefficient of determination; Skew—skewedness; Kurt—kurtosis; StDev—standard deviation; Var—variance.

**Table 9 antibiotics-12-00103-t009:** Product specification of essential oils.

Essential Oil	Lemongrass	Oregano	Rosemary	Peppermint	Eucalyptus
Product INCI	*Cymbopogon citratus* leaf oil	*Origanum vulgare* oil	*Rosmarinus officinalis* leaf oil	*Mentha piperita* oil	*Eucalyptus globulus* leaf oil
CAS Number	8007-02-1	84012-24-8	8000-25-7	84082-70-2	84625-32-1

## Data Availability

Not applicable.

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
