# Peer review of "Screening of Antifungal Activity of Essential Oils in Controlling Biocontamination of Historical Papers in Archives"

_antibiotics, 2023, doi:10.3390/antibiotics12010103_

Round 1

Reviewer 1 Report

The study by on the screening and EOs with antifungal activity for controlling selected Paper fungi is interesting and suitable for antibiotics Journal. The experiment is robust, but the discussion is lacking. Some comments will help improve the MS. Also, pls check the attached pdf for highlights and comments

-       The abstract is confusing and not informative enough. This abstract needs a rework. Important results (not necessarily numeric data) should be informed. In addition, it contains a lot of grammatical errors. Also, many important pieces of information are missing.

-       Why was MIC presented in percentage (%)?

-       Is there a basis for selecting the 5 EOs screened in this study?

-       Add the unit for inhibition zones in Table 1

-       L123 – Dilutation -> Dilution

-       Table 2 lacks replication.

-       Please provide more explanation on the enhanced antifungal effects of the MI, as observed for A. fumigatus and P. chrysogenum (Line 118-121)

-      How many Petri plates were done in Table 6? Add EO concentration used?

-      Table 8 is confusing. Drag the column so that all values can fit in a horizontal position.

-       Highlights in the pdf attached indicate a few grammatical errors. A thorough English editing is necessary for the publication of this manuscript

-       Table 9. Authors should specify CAS based on the extract used in this study. For example, Rosemary 8000-25-7 is oil, while 84604-14-8 is a powdered extract.

-      L360 – Please explain “freshly prepared overnight fungal culture”

-      L366-367 Can this be added to Table 1 instead

-      Eqn 1:- Change to formulae

-       L 387- 120 h

-       Since the mixture was found more effective, it would have been very interesting to equally investigate the inhibition effects of the EO Mix in different ratios (equal ratio 1:1:1,v/v/v was used in this study).

Major Comments

Antibiotics journal allows a free-format form of MS. But, when the result and discussion come before M&M, merging the Result with its discussion is recommended. The major flaw will be difficulty in understanding this study. The information is too disjointed, and the results were not even well discussed in Section 3. Authors should consider this suggestion.

Throughout the discussion, the authors did not attribute the observed effects to a specific component(s) of the EO, which I believe could be corroborated by the literature. The discussion is L 289 to 293 is insufficient

Pdf attached

Author Response

Please, find the answers in the word file

Reviewer 2 Report

In the paper entitled “Screening of antifungal activity of essential oils in controlling biocontamination of historical papers in archives”, the antimicrobial effects of five plant essential oils including lemongrass, oregano, rosemary, peppermint, and eucalyptus were investigated for the protection of historical papers in archives. It was found the mixture of lemongrass, oregano, and peppermint essential oils displayed good inhibition effect. This work could provide a theoretical and experimental bases for the discovery and development of new historical paper protection agent candidates. I recommend this manuscript be accepted.

Author Response

(The authors gave the same response as above.)

Reviewer 3 Report

The manuscript “Screening of antifungal activity of essential oils in controlling biocontamination of historical papers in archives” is written in clear and concise language that helps the reader stay interested.

In addition to what has been mentioned, the authors explain in detail the use of these compounds and the results they have had through the different experiments carried out. From these results, the authors wrote a very comprehensive discussion, explaining interesting details such as the variation that the chemical composition of the EOs used may have depending on their origin. Also, the references used are extensive and up-to-date.

Below, I suggested some modifications that could help authors improve their manuscript:

Figures 1 and 4 have quality problems, at least in the pdf that I have been able to download from the web. Could it be improved to show better image quality? In addition, a bracket is missing in the information (Line 131-132).

Line 120 and 212: I think you meant to write EOs instead of Eos.

Table 3: Capitalization should be used for Oregano, Peppermint and Lemongrass as you have done in the rest of the tables.

Line 242: I guess y Capitalization should be used “L”emongrass

Line 359-361: As a suggestion, I think the authors should include how they prepared the overnight culture or give more information about it.

Line 405: 205ºC needs separation.

Author Response

(The authors gave the same response as above.)

Round 2

Reviewer 1 Report

The manuscript has improved. However, some sentences are difficult to read or understand. I must insist on language editing.

A few;

- Abstract: L28-31, L32-34.

- For MIC, is it % w/v or % v/v? Clarify

- Check Table 8 again. The changes - 0.005 looks like the dash and values are separated